# Transcriptomic and Metabolomic Profile Analysis of Muscles Reveals Pathways and Biomarkers Involved in Flavor Differences between Caged and Cage-Free Chickens

**DOI:** 10.3390/foods11182890

**Published:** 2022-09-17

**Authors:** Liubin Yang, Fang Yuan, Li Rong, Jinping Cai, Sendong Yang, Zijia Jia, Shijun Li

**Affiliations:** 1College of Food Sciences & Technology (Ministry of Education), Huazhong Agricultural University, Wuhan 430070, China; 2Key Laboratory of Agricultural Animal Genetics, Breeding and Reproduction, Ministry of Education, Huazhong Agricultural University, Wuhan 430070, China; 3Key Laboratory of Smart Farming for Agricultural Animals, Ministry of Education, Huazhong Agricultural University, Wuhan 430070, China

**Keywords:** native chicken breeds, exercise, RNA-seq, raw meat, flavor precursors

## Abstract

The cage-free system has gained a lot of interest in recent years because it can offer chickens more freedom and is easier to manage compared with free-range rearing systems, but few studies have focused on the effect of the cage-free rearing system on meat quality and flavor. In this study, 44 Jianghan chickens were reared in caged or cage-free systems to explore the effect of different rearing systems on meat-eating quality. Sensory evaluation of cooked muscles showed that the leg muscle aroma, juiciness, and flavor intensity significantly improved by the cage-free rearing. The cage-free hens had significantly lower body weight, abdominal fat percentage, and meat fat content, but higher meat moisture content. The cage-free group had brighter breast muscle and redder leg muscle color 24 h after slaughter. Transcriptomic and metabolomic profile analysis of the leg muscle samples showed that the cage-free rearing changed biosynthesis pathways associated with glycogen metabolism, lipid and fatty acid biosynthesis and transport, muscle cellular type, and cellular components, which were related to raw meat quality. Different rearing systems also resulted in differences in glycolipid metabolism, lipid metabolism, and altered levels of intramuscular fat content and other flavor precursors. Pathways such as glycerolipid metabolism, adipocytokine signaling, and metabonomic pathways such as linoleic acid, glycerophospholipid, arginine, proline, and β-alanine metabolism may be responsible for the meat quality and flavor change.

## 1. Introduction

China is among the top poultry consumers and producers in the world. The consumption of poultry products increases yearly. There are hundreds of native chicken breeds in China, which greatly satisfies the consumers’ pursuit of chicken meat [1]. Unlike the imported fast-growing broiler or commercial layer breeds, most Chinese native chickens were used for both egg and meat producing purpose. For thousands of years, these native chickens were dispersed in small scales and reared in the free-range system. Due to the low production capacity of free-range, it is only suitable for family farming [2]. Over the past 40 years, new chicken rearing systems have been rapidly introduced. Most native chicken rearing has been transferred from the traditional free-range to cage rearing system as the production has expanded. However, due to the lack of exercise, caged chickens have lower disease resistance and high abdominal fat, and consumers often complain that the eating quality is not as good as that of free-range chickens [3,4].

Meat-eating quality is the dominant factor influencing consumers’ choice, which depends on comprehensive sensory attributes, including meat flavor, tenderness, and juiciness [5]. Among these characteristics, meat flavor depends on both water- and fat-soluble compounds in meat. Meat flavor can be generated through various reactions, such as Maillard reaction, lipid oxidation, vitamin degradation, and the interactions with different components and precursors during the cooking process. Sugars, nucleotides, free amino acids, peptides, lipids, fatty acids, and thiamine are the major flavor precursors in raw meat [6]. Animal breed, sex, age, feed, chiller aging, precursor content, and cooking method are also factors that affect meat flavor [5]. Meat texture, especially tenderness, is another important aspect of meat quality. External factors (such as nutritional status and post-slaughter treatment) together with internal factors (muscle fiber type, composition, glycolysis, calcium release, and protease activation) all contribute to variations in meat tenderness [7,8].

The cage-free system has gained a lot of interest in recent years because it can offer chickens more freedom compared with the caged system and is easier to manage compared with the free-range system. The cage-free system can lead to better animal welfare. Chickens rearing in a cage-free system are more active, naturally behaved, and have a better resistance to disease [9,10,11]. It is also reported that rearing systems have great impacts on chicken growth, carcass yield, egg quality, and meat production [12,13,14].

Although the external factors (such as environment, rearing system, and feed) cannot change the genotype of chickens, their effect on animal economic traits cannot be ignored. Moreover, the epigenome comprising different mechanisms, e.g., DNA methylation, remodeling, histone tail modifications, chromatin microRNAs, and long non-coding RNAs, interact with environmental factors such as nutrition, pathogens, and climate to influence the expression profile of genes and the emergence of specific phenotypes [15,16]. Multi-level interactions between the genome, epigenome, and environmental factors might occur [17]. Furthermore, numerous lines of evidence suggest the influence of epigenome variation on health and production [15,18,19]. The expression of eukaryotic genes is temporarily and multidimensionally controlled [20]. Only a relatively small set of the entire genome is expressed in each type of tissue, and the expression of genes depends on the stage of development [21]. Therefore, gene expression in eukaryotes is specific to each tissue [22]. In addition, the amount of gene products that are made in the same tissue, as well as in other tissues that make up that product, regulates the expression of that gene [23]. One of the basic activities in domestic animals is the study of genes and proteins related to economic traits and their study at the cellular or chromosomal level [24]. It has been reported that the cage-free chickens showed different metabolic and physiological characteristics in egg fatty acids content, and gene expression patterns in the oviductal magnum and gut microbiome compared with caged ones [25,26,27]. However, there is a lack of studies focusing on the impact of a cage-free system on the specific metabolites of chicken meat and the links between muscle biosynthesis pathways and the flavor profiles of meat. 

Foodomics, including transcriptomics, metabolomics, and lipidomics, have been widely used to reveal physiological and biochemical changes and flavor precursors in food materials to improve our understanding of flavor precursor composition and flavor formation [17,18,19,20]. In this study, one of the Chinese famous native, dual-purpose breeds “Jianghan” chickens (eliminate at 60~75 weeks of age) was used as materials [28]. Comparative transcriptomic and untargeted metabolomic techniques were employed to investigate the flavor-related metabolic differences between caged and cage-free hens. The aim of this study was to investigate the effect of different rearing systems on chicken-meat-eating quality and the mechanism responsible for it.

## 2. Materials and Methods

### 2.1. Ethics Approval Statement

All animal experiments were carried out according to protocols (No. 5 proclaim of the Standing Committee of Hubei People’s Congress) approved by the Standing Committee of Hubei People’s Congress, and the ethics committee of Huazhong Agricultural University, China. 

### 2.2. Animal Resources and Rearing Systems

Before the experiment, all Jianghan hens were reared in individual stair-step cages (with a size of 47 × 37 × 33 cm) from 17 to 60 weeks of age at Huazhong Agricultural University chicken farm. Chickens were under standard management with a suitable environment, lighting regime (16 h light: 8 h dark), sufficient feed (Nongteng-304, China), and water. After 60 weeks, 44 hens were randomly separated into caged and cage-free groups. The caged group was maintained in the same cages as used before the experiment, while the cage-free group was placed collectively in a rearing room with an area of approximately 40 m^2^. All the other management and feeding practices were identical for both groups. The experiment lasted for 2.5 months [13,29].

### 2.3. Carcass Measurement and Meat Quality Assessment of Hens from Different Rearing Systems

Before and after the experiment, the body weight of each chicken was recorded. After slaughter, the abdominal fat of each hen was measured. Then, the carcass was separated into two parts. The right part of the carcass was used to measure the basic parameters of meat (each group of twenty-two samples). A Raman spectrum analyzer (Food Scan, Beijing, China) was used to identify the protein, moisture, fat, and collagen contents of total meat (mixture of boneless chest muscle and leg muscle). Parameters including muscle color, pH, and water holding capacity (WHC) were used for muscle quality assessment (each group of twenty-two samples). A colorimeter (3nh-NR100, Shenzhen, China) was used to measure the color of breast and leg muscles at 45 min and 24 h after slaughter. Muscle color assessment used the “a^#^ (red/green)”, “b^#^ (blue/yellow)”, and “L (lightness)” values. A pH meter (Qiwei E201-C, Hangzhou, China) was used to measure the pH value of leg muscle and breast muscle 45 min and 24 h after slaughter. The Kauffman method was used to assess the water holding capacity (WHC) of leg muscles and breast muscles [30].

### 2.4. Sample Preparation

The left part of the carcass was used for morphological observation, RNA-sequencing (RNA-seq), and sensory evaluation. Specifically, the leg muscles were taken from the same position from the iliotibial lateral muscle, and the breast muscle samples were sampled from the same position at the center of the breast muscle. One half of the muscle samples were used for optimal cutting temperature compound (OCT) embedding and paraformaldehyde fixation. The other half of the samples were homogenated with Trizol reagent (Beyotime, Shanghai, China) and stored in liquid nitrogen until use. All samples were collected within 5 min after slaughter. After sampling, the remaining parts were stored at −20 °C for sensory evaluation.

### 2.5. Sensory Evaluation

Leg and breast muscles from each treatment group were prepared separately for sensory evaluation. Briefly, each sample was put on a deep plate and then steamed at 100 °C for 2 h using a steamer. The steamed sample was cut into 2~3 pieces (2 × 1 × 1.5 cm meat chunks). A total of 41 volunteers participated in the sensory evaluation. Panelists’ age ranged from 21 to 36, and the gender ratio was about 2:1 (female:male). All panelists consumed chicken meat regularly in their daily life. All participants were taught how to infer scores before the test. Before tasting, samples were cooled to 40 °C, and each volunteer was randomly served with one chunk of leg meat and one chunk of breast meat separately for evaluation. An 8-point scale (1 to 8) was used to evaluate the intensity of aroma, initial impression of juiciness, first bite, sustained impression of juiciness, muscle fiber and overall tenderness, amount of connective tissue, overall flavor intensity, and off flavor intensity. The detailed method of sensory evaluation and statistical analysis followed similar steps in a previous study [31].

### 2.6. Morphological Studies of Leg Muscle Samples

A total of eight leg muscle samples (each group of four samples) were used for morphological evaluations. Tissues were embedded perpendicular to the muscle bundle. Morphological differences were observed by paraffin sections with hematoxylin-eosin staining. Intermuscular fat (IMF) was identified by frozen sections and oil red O staining. Hematoxylin-eosin staining and oil red O staining of sections were performed according to the same steps as used in our previous study [32].

### 2.7. RNA-Seq and Data Analysis and qRT–PCR Verification

RNA was isolated using the phenol/chloroform RNA-extraction method [33]. After RNA isolation, 10 samples (each group of 5 samples) were sent to the Majorbio biotechnology company (Shanghai, China) for paired-end 150 mRNA sequencing. Each sample produced more than 6 GB of clean data. Data analysis followed the same steps as in our previous study [25]. The cage-free group was compared to the caged group to reveal differentially expressed genes (DEGs) and pathways between different rearing systems. Genes with *p* value < 0.05 and |log2(Foldchange)| > 1 were considered differentially expressed genes (DEGs). Gene Ontology (GO) enrichment and Kyoto Encyclopedia of Genes and Genomes (KEGG) analysis of DEGs were conducted by the KOBAS 3.0 website (http://kobas.cbi.pku.edu.cn/kobas3/, accessed on 16 April 2021). Terms with a *p* value < 0.05 were regarded as significantly enriched.

Candidate genes, including *COL3A1 FABP3*, *FBN1*, *GLS2,* and *NR4A3*, were selected to verify the reliability of RNA-seq. cDNA was synthesized using the one-step gDNA removal and cDNA synthesis supermix (TransScript, Beijing, China). A total of 10 samples (5 from caged groups; 5 from cage-free groups) were used for qRT–PCR. The primers were designed by Oligo 7.6 (Molecular Biology Insights, Colorado Springs, CO, USA). Primer information is listed in Appendix A. The quantitative real-time PCR (qPCR) amplification conditions were as follows: 95 °C for 10 min; 40 cycles of 95 °C for 15 s and 60 °C for 1 min; 95 °C for 15 s; 60 °C for 1 min; 95 °C for 1 s. The fold change was calculated by the 2^−ΔΔCt^ method followed by significance testing by *t* test.

### 2.8. Untargeted Metabolomics and Data Analysis

Eighteen leg muscle samples (2 treatments × 9 biological replicates) were prepared for untargeted metabolomics analysis by Liquid Chromatography–Mass Spectrometry (LC–MS) (Thermo, uhplc-q exactive HF-X). Metabolite extraction and quality control were performed by the standard sample preparation process of Majorbio Biotechnology Company (Shanghai, China). The chromatographic column was ACQUITY UPLC HSS T3 (100 mm × 2.1 mm I.D., 1.8 µm; Waters, Milford, MA, USA). Mobile phase A was 95% water with 5% acetonitrile (containing 0.1% formic acid), and mobile phase B was 47.5% acetonitrile with 47.5% isopropyl alcohol and 5% water (containing 0.1% formic acid). The sample injection volume was 2 µL, and the flow rate was set to 0.4 mL/min. The column temperature was maintained at 40 °C. Quality control, missing value recoding, and data normalization were conducted using the Majorbio standard process.

The cage-free group was compared to the caged group to reveal differential metabolites between the two rearing systems. Orthogonal partial least squares discriminate analysis (OPLS-DA) was used for statistical analysis to determine global metabolic changes between comparable groups. Metabolites with VIP values > 1 and *p* values < 0.05 were considered differentially abundant. GO enrichment and KEGG analysis of differentially accumulated metabolites were conducted using the Majorbio Cloud Platform (http://www.majorbio.com/, accessed on 12 March 2021). Terms with a *p* value < 0.05 were regarded as significantly enriched.

### 2.9. Comparative Analysis of RNA-Seq and Untargeted Metabolomics Data

The correlation between RNA-seq and untargeted metabolomics data was assessed by the Omicshare website O2PLS online tools (https://www.omicshare.com/tools/Home/Soft/o2pls, accessed on 18 August 2021). Correlations between DEGs and differential metabolites were analyzed by the Metaboanalyst 5 online tools (https://www.metaboanalyst.ca/, accessed on 20 October 2021).

## 3. Results

### 3.1. Carcass and Meat Quality under Different Rearing Systems

Statistical analysis of carcass and meat quality parameters is shown in Table 1. The body weight of cage-free hens was decreased, while the body weight of caged hens was increased compared with the beginning of the experiment. The abdominal fat of hens in the cage-free groups was 2.49%, which was significantly lower than that in the caged group (4.65%). The meat content analysis showed that samples from the cage-free group showed a significant increase in moisture content and a decrease in fat content, but no difference in protein and collagen content compared to the caged group. Meat quality assessment of muscles revealed that cage-free rearing significantly decreased leg muscle WHC, but no effect was found on breast muscle WHC. Both breast and leg muscle pH increased in cage-free groups 45 min after slaughter, but there was no difference between the two groups 24 h after slaughter. The breast muscle color a^#^ and b^#^ values in the cage-free group showed a significant decrease 45 min after slaughter, the a^#^ value showed a continuous decline, whereas the L value significantly increased 24 h after slaughter. Only leg muscle color a^#^ in cage-free groups showed a significant increase 45 min after slaughter, but all these values showed significant differences compared with the caged group 24 h after slaughter.

### 3.2. Sensory Evaluation of Cooked Meat

Leg muscles showed higher sensory characteristics than those of chest muscles in general except for the amount of connective tissue. The performance of sensory panelists was first evaluated and the results showed that age had no significant (*p* > 0.05) effect on the rating scores (Appendix A). Gender only significantly affected (*p* < 0.05) the overall sensory rating of leg muscle, of which females tended to give a higher score than males possibly because women tend to be more reactive to stimuli [34], and possibly because the females were more sensitive to the flavor. Overall, the individual differences were negligible, so the rating scores were further used to evaluate the effect of the rearing system on chicken meat sensory attributes. The results showed that the free-cage rearing significantly (*p* < 0.05) improved leg muscle aroma intensity, sustained impression of juiciness, overall flavor intensity scores, as well as breast muscle fiber and overall tenderness score (Figure 1A).

### 3.3. Meat Morphological Changes under Different Rearing Systems

Morphological observation showed that both meat morphological characteristics and IMF deposition were different between the two rearing groups. The muscle fibers of cage-free hens were obviously looser, and there was more IMF deposited between the muscle bundles compared to the caged group (Figure 1B–E).

### 3.4. RNA-Seq and Differentially Expressed Gene Analysis

Quality control analysis suggested that the quantity and quality of RNA-seq clean data were good enough for the following gene expression analysis. The cage-free group was compared to the caged group to investigate the DEGs and pathways that were related to the different rearing systems. Pearson correlation analysis showed there is a high repeatability of samples within groups (Figure 2A). A total of 745 DEGs were identified (Appendix A), among which 239 were upregulated and 506 were downregulated (Figure 2B). qPCR verification showed that all five selected genes followed a similar expression tendency with the RNA-seq data (Figure 2C). The gene expression heatmap of the top 200 DEGs showed that the gene expression patterns were significantly different between the two groups (Figure 2D).

A total of 152 GO terms and 6 pathways were significantly enriched. The top 20 enriched GO terms of DEGs are shown in Figure 2E. GO terms such as glycogen metabolic process (4 genes were enriched, including *PRKAG2*, *UGP2*, *LEPR*, and *PHKA2*), fatty acid oxidation (3 genes were enriched, including *LEPR*, *NR4A3*, and *C1QTNF2*) and terms such as glycolipid metabolism (7 genes were enriched, including genes such as *PNPLA2*, *MBOAT2*, and *LPIN1*), and adipocytokine signaling pathway (6 genes were enriched, including genes such as *TRADD*, *PRKAG2*, *CPT1A*, *LEPR*, and *MAPK9*) were included. Significantly enriched KEGG pathways and GO terms that were potentially related to muscle quality or flavor changes are shown in Table 2 and Table 3, respectively.

### 3.5. Untargeted Metabolomics and Different Metabolite Analyses

Quality control analysis suggested that the quantity and quality of metabolome assay data were good enough for the following metabolite expression analysis. The OPLS-DA showed that there were significant differences between the cage-free and caged groups (Figure 3A,B). The cage-free group was compared with the caged group to investigate the differential metabolites and pathways. A total of 408 differential metabolites (Appendix A) were identified, among which 173 were upregulated and 271 were downregulated (Figure 3C). Metabolites that are potentially related to meat flavor and flavor precursors are shown in Table 4. The cluster analysis of the top 200 differential metabolites is shown in Figure 3D. KEGG pathway enrichment of differential metabolites showed 15 significant pathways. Pathways such as lipid metabolism and metabolism of other amino acids are included (Figure 3E).

### 3.6. Comparative Analysis of RNA-Seq and Untargeted Metabolomics Data

Comparative analysis between RNA-seq and untargeted metabolomics data showed that they followed similar expression patterns. The top 30 DEGs and differential metabolites are shown in Figure 4A. Genes and metabolites involved in glycolipid metabolism and glycophospholipid metabolism showed the closest relationship between the transcriptome and metabolism (Figure 4B).

## 4. Discussion

In this study, we mainly focused on the impact of different rearing systems on the chicken-meat-eating quality with an emphasis on the flavor-related characteristics. We did not separate the chickens into different rearing systems from chicks or young chickens. The main reason was that the difference in growth and egg laying process may also be affected by rearing systems [12,29], which makes the results more complex and difficult to interpret. More importantly, at the late laying stage, the laying rate was significantly decreased and hens tended to become fat, with a low immunity due to lack of exercise [3,35]. Therefore, the chickens were separated into the caged and cage-free groups at this stage to increase the activity of chickens, improve animal welfare, promote body health, and obtain meat products with better quality and flavor before the slaughter. 

The cage-free hens showed more natural behaviors, such as jumping, fly, fighting, perch, and preening, and had more physical activity than the caged group during the experiment. Statistical results showed that cage-free rearing had different effects on carcass and meat-quality-related parameters. The body weight and abdominal fat percentage were significantly decreased in the cage-free rearing group, which proved that physical metabolism could be affected by different rearing systems. Although the fat content decreased in whole meat, morphological evaluation showed a contrary increase in IFMs in cage-free chicken muscles, which provides more evidence of the physical metabolism changes, specifically for the fat metabolism. Cage-free rearing affects muscle quality in many distinct aspects, among which muscle color and pH seem closely related to the measurement time point after slaughter. The collagen content showed a strong correlation with WHC and pH [36]; however, our results found no differences of it between different groups. This suggested that other factors may be responsible for the WHC and pH differences in different rearing systems. It is noteworthy that the cage-free group had higher moisture content and lower fat content in whole meat, lower WHC in leg muscles, brighter breast muscle color, and redder leg muscle color, all of which were considered to be beneficial to meat quality [9,12,19,37] []. These results were consistent with the sensory evaluation results that after cooking, the leg muscles from the cage-free group were juicier and tender due to the higher WHC and looser muscle bundles. IMF is rich in unsaturated fatty acids, which were important meat flavor precursors. Additionally, it also serves as an important solvent for various flavor substances [6,38,39]. More IMF deposited between the muscle bundles may be one of the main reasons for a better flavor sensory of cage-free groups. 

In this study, we also found that different rearing systems had more impact on the sensory attributes of leg muscles than breast muscles, possibly because the leg muscles were crucial tissue used in walking, jumping, fighting, and physical exercise. In order to further understand the effect of the rearing system on the biosynthesis pathways of flavor-related compounds, the leg muscles were used for further transcriptomic and metabolomic analysis.

In humans, physical exercise improves health and leads to the remodeling of skeletal muscle phenotype reflected by changes in myogenesis, glucose metabolism, lipid metabolism, angiogenesis, transcriptional regulation, and mitochondrial genes [40,41,42]. Similarly, transcriptomic and metabolomic changes were also observed in hens under cage-free rearing compared with caged hens. Transcriptomic analysis between the two groups revealed a series of DEGs potentially responsible for meat quality. For example, *MYPN* was downregulated in the cage-free group, which was reported to be specifically expressed in striated muscle and played a critical role in the control of muscle growth [43]. *PDLIM3* was also upregulated in the cage-free group, which regulates the proliferation and differentiation of skeletal muscle satellite cells [44,45]. As the main component of myointima, collagen-encoding and collagen-binding genes, such as *COL6A1*, *COL6A2*, *COL6A3*, and *VCL*, were identified as downregulated DEGs. *UGP2*, *PHKA2*, and *LEPR* are genes important for regulating glycogen metabolic patterns [46,47,48], and the upregulation of *UGP2* and *LEPR* and the downregulation of *PHKA2* suggested a novel glycogen metabolic pattern in muscle. In addition, a series of genes related to oxygen carrier activity, oxygen binding, hydrogen peroxide catalytic process, and peroxidase activity were identified as DEGs. The results clearly showed that physical exercise promotes muscle remodeling and stimulates muscle satellite cell activation to repair damaged cells. Our results suggested that the muscle cell type, cellular components, and metabolic pattern were positively affected by cage-free rearing, leading to changes in meat color, pH, and WHC, which could be related to the tenderness and juiciness of the meat.

The generation of cooked meat flavor mainly depends on the content of precursors [5,6]. Both RNA-seq and metabolomics revealed that flavor precursors, including lipids, fatty acids, free amino acids, sugars, nucleotides, free amino acids, peptides, and thiamine, were affected by the rearing systems. Changes involved in lipid and fatty acid metabolism were the most noteworthy in this study. Cage-free rearing reduced the abdominal fat and fat content (mainly intermuscular fat) in meat but increased the muscle IMF content. A series of genes participating in lipid and fatty acid metabolism were identified. For instance, *NR4A3* was regarded as one of the most exercise- and inactivity-responsive genes identified in a meta-analysis and were involved in the regulation of glucose and fatty acid utilization in skeletal muscle [49,50,51]. *MFSD2A* was reported to be an important lipid transporter that was essential for omega-3 fatty acid docosahexaenoic acid transport [52,53]. *FABP3* belonged to the fatty acid-binding protein family, which was highly expressed in skeletal muscles and served as a lipid “chaperone” in regulating fatty acid metabolism [54,55]. The *PNPLA2* gene encoded an enzyme that catalyzed the first step in the hydrolysis of triglycerides in adipose tissue [56]. The upregulation of *NRA43*, *MFSD2A*, and *FABP3* and the downregulation of *PNPLA2* suggested changes in lipid and fatty acid synthesis, transportation, and metabolism in muscles due to cage-free rearing. Metabolomic analysis revealed a series of lipids and lipid-like metabolites, such as LysoPC (22:4(7Z,10Z,13Z,16Z)), glycerophosphocholine, and succinylcarnitine. Linoleic acid metabolism and glycerophospholipid metabolism were the most enriched pathways. Comparative analysis confirmed that the genes and metabolites related to lipid metabolism were highly correlated in this study. It is well known that fat content, especially IMF content, plays a crucial role in meat tenderness, juiciness, and flavor [6,38,39]. Therefore, the changes in lipid and fatty acid metabolism, especially the IMF content change in muscles, could contribute to eating quality differences between different rearing systems.

Metabolomic analysis revealed that arginine, L-proline, and β-alanine metabolism were affected by the different rearing systems. Among these compounds, arginine and L-proline are regarded as important flavor compounds in puffer fish meat [57,58] and were more abundant in the cage-free group. Carnosine is widely distributed in skeletal muscle; plays roles in antioxidant capacity, pH buffering, and metal-ion chelation; has a close relationship with glycolytic metabolism in muscle [59,60]. Glutathione plays an essential role in scavenging radicals, detoxification, and signal transduction. These differential amino acids and peptides not only function as flavor precursors in meat but also participate in the metabolic process in muscle [61,62]. Additionally, some nucleosides, nucleotides, vitamins, and other compounds, such as 5’-CMP, DIMP, pantothenic acid, choline compounds, and alcohol compounds, were also identified as differentially metabolized in this study and might be potential flavor precursors in raw meat.

## 5. Conclusions

In conclusion, cage-free rearing significantly reduced body weight and abdominal fat percentage, showed different impacts on the meat-quality-related parameters, and helped to improve sensory evaluation scores of muscles. Transcriptomic analysis revealed genes and pathways related to meat quality and flavor changes. Metabolite analysis showed that lipid metabolism, intramuscular fat, and other flavor precursor metabolisms were affected by the rearing system, which had a potential impact on meat flavor. Overall, the cage-free system showed a positive effect on the improvement of chicken-muscle-eating quality.

## Figures and Tables

**Figure 1 foods-11-02890-f001:**
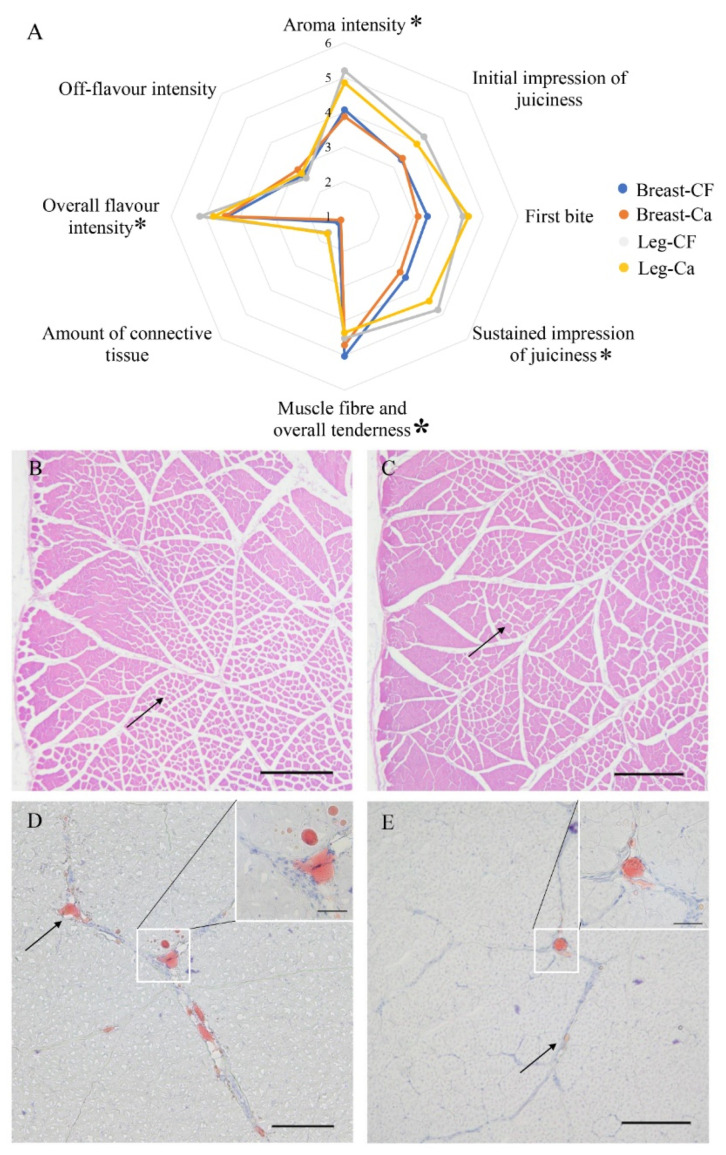
(**A**) Statistical analysis of sensory evaluation from different groups. Different colors refer to different parts of muscles and groups. “Breast” refers to breast muscle, “Leg” refers to leg muscle, “CF” refers to the cage-free group, and “Ca” refers to the cage-rearing group. The free-cage rearing significantly improved scores of aroma intensity, sustained impression of juiciness, overall flavor intensity of leg muscles (marked with “*”), and the score of muscle fiber and overall tenderness of breast muscles (marked with bold “*”). Morphological differences in leg muscles between the cage-free and caged groups. The paraffin section and hematoxylin-eosin (H&E) staining showed that the muscle fibers of the cage-free group (**B**) were looser than those of the caged group (**C**). Oil red O staining showed that there was more intramuscular fat deposited between the muscle bundles in the cage-free group (**D**) than in the caged group (**E**). The fat was stained red and indicated by the arrow. Normal scale bar, 100 μm; enlarged scale bar, 20 μm.

**Figure 2 foods-11-02890-f002:**
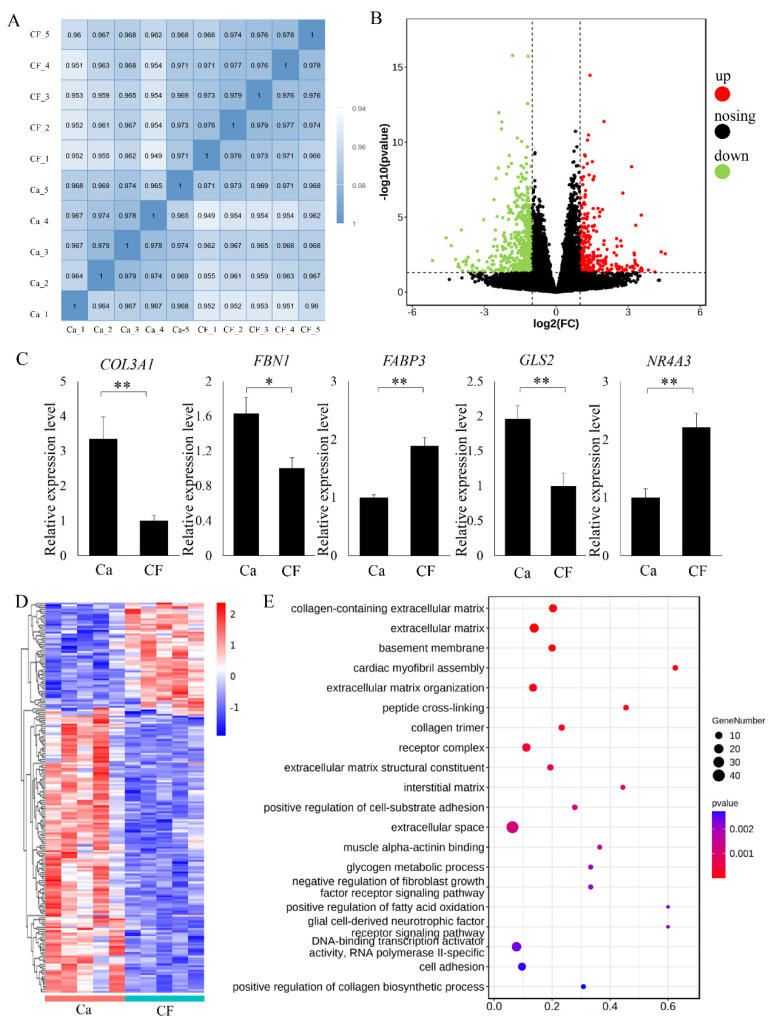
Transcriptome analysis of leg muscle samples from cage-free and caged groups. “Ca” refers to the caged rearing group, and “CF” refers to the cage-free group. (**A**) Pearson correlation analysis between samples. “CF” refers to the cage-free group, and “Ca” refers to the cage-rearing group. Numbers refer to correlation coefficient between samples. (**B**) Volcano plot of differentially expressed genes. The X-axis refers to the log2-fold fold change, and the Y-axis refers to −log10 (*p* value). The red dots and green dots represent the upregulated and downregulated genes, respectively. (**C**) qPCR verification of RNA-seq results. All 5 genes (*COL3A1*, *FBN1*, *FABP3*, *GLS2*, and *NR4A3*) followed a similar RNA-seq expression pattern. The X-axis represents different developmental groups, and the Y-axis represents the relative expression levels. “*” and “**” refer to significant differences with *p* values < 0.05 and < 0.01, respectively. Error bars indicate standard errors. (**D**) Heatmap of the top 200 DEGs. Different colors refer to different expression levels. (**E**) The top 20 enriched GO terms of DEGs. The X-axis refers to the rich factor. The Y axis refers to different GO terms.

**Figure 3 foods-11-02890-f003:**
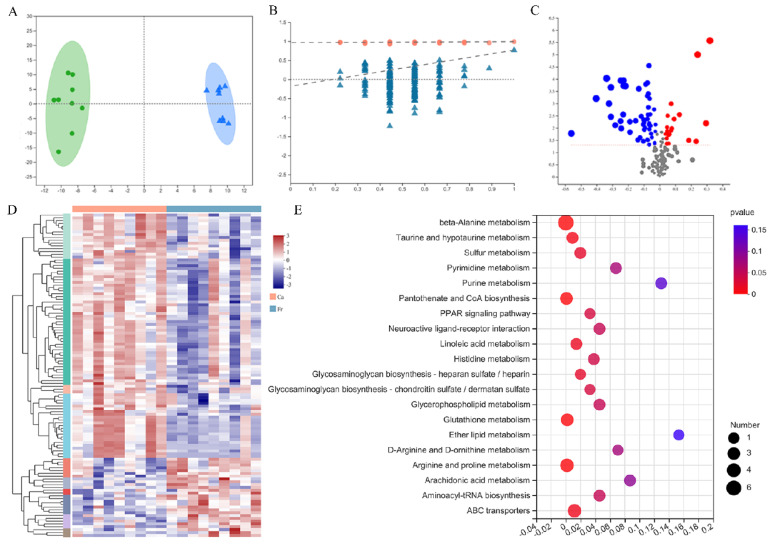
Metabolome analysis of leg muscle from different rearing conditions. “Ca” refers to the cage rearing group, and “Fr” refers to the cage-free rearing group. (**A**) OPLS-DA score plot of metabolites in leg muscle samples. “Dot” refers to the cage rearing group, and “triangle” refers to the cage-free rearing group. (**B**) The permutation tests of data; the Y axis refers to similarity. (**C**) Volcano plot of differentially expressed metabolites. The X-axis refers to the log2-fold change, and the Y-axis refers to −log10 (*p* value). The red dots and blue dots represent the upregulated and downregulated metabolites, respectively. (**D**) The heatmap of the top 200 differential metabolites. Different colors refer to different expression levels. (**E**) The KEGG pathway analysis of differential metabolites. The X-axis refers to the *p* value and the Y-axis refers to different pathways.

**Figure 4 foods-11-02890-f004:**
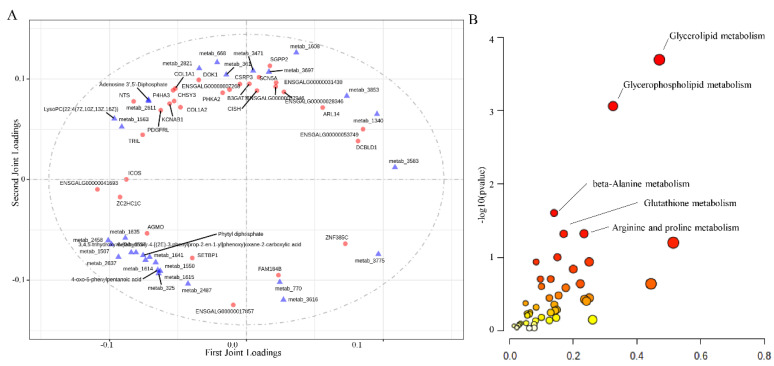
Comparative analysis between transcriptome and metabolome data. (**A**) Comparative analysis between RNA-seq and metabolome data. The X and Y axes refer to the first joint loading and second joint loading, respectively. The absolute value of the number in coordinates indicates the degree of correlation of the two omics. The red dot refers to the gene, and the blue triangle refers to the metabolite. (**B**) Comparative analysis between DEGs and differential metabolites. The X-axis refers to the pathway impact between the two omics. The y-axis refers to −log10 (*p* value).

**Table 1 foods-11-02890-t001:** Comparison of carcass and meat quality parameters between different rearing conditions.

Traits	Value	
	Ca	CF
Body weight change (g)	127 ± 153	−132 ± 115 ***
Abdominal fat rate (%)	4.65 ± 1.48	2.49 ± 0.98 ***
Breast muscle WHC (%)	11.81 ± 6.08	12.26 ± 8.17
Leg muscle WHC (%)	12.74 ± 4.97	8.22 ± 2.37 ***
Meat moisture content (%)	73.33 ± 0.56	74.19 ± 0.54 ***
Meat protein content (%)	22.23 ± 0.48	22.06 ± 0.48
Meat fat content (%)	3.68 ± 0.68	3.22 ± 0.45 *
Meat collagen content (%)	1.56 ± 0.09	1.61 ± 0.10
Breast muscle pH (45 min)	5.07 ± 0.32	5.88 ± 0.14 ***
Breast muscle pH (24 h)	5.69 ± 0.09	5.79 ± 0.26
Leg muscle pH (45 min)	5.30 ± 0.19	6.09 ± 0.15 ***
Leg muscle pH (24 h)	5.94 ± 0.13	5.91 ± 0.21
Breast muscle color a^#^ (45 min)	9.92 ± 1.07	5.25 ± 1.39 ***
Breast muscle color b^#^ (45 min)	6.40 ± 1.24	4.42 ± 1.35 ***
Breast muscle color L (45 min)	48.13 ± 4.57	50.18 ± 2.80
Breast muscle color a^#^ (24 h)	8.88 ± 1.17	4.79 ± 2.00 ***
Breast muscle color b^#^ (24 h)	5.87 ± 2.37	5.57 ± 1.59
Breast muscle color L (24 h)	52.13 ± 3.41	58.04 ± 3.73 ***
Leg muscle color a^#^ (45 min)	4.91 ± 1.25	10.04 ± 1.67 ***
Leg muscle color b^#^ (45 min)	4.83 ± 1.71	5.18 ± 1.45
Leg muscle color L (45 min)	49.67 ± 3.31	50.12 ± 4.95
Leg muscle color a^#^ (24 h)	4.80 ± 1.64	11.35 ± 9.78 *
Leg muscle color b^#^ (24 h)	6.63 ± 2.10	4.94 ± 1.64 **
Leg muscle color L (24 h)	60.64 ± 3.29	53.33 ± 3.05 ***

“Ca” refers to the caged system, and “CF” refers to the cage-free system. “*” refers to *p* value < 0.05, “**” refers to *p* value < 0.01, and “***” refers to *p* value < 0.001. Muscle color a^#^ refers to the red/green value. “b^#^” refers to the blue/yellow value. L refers to lightness.

**Table 2 foods-11-02890-t002:** Significantly enriched KEGG pathways are potentially related to muscle quality and meat flavor.

Term	Count	Gene
ECM-receptor interaction	11	*COL1A2*, *COL6A1*, *COL6A2*, *COL6A3*, *ENSGALG00000042388*, *FN1, LAMC1*, *NPNT*, *RELN*, *THBS1*, *THBS2*
Glycerolipid metabolism	7	*AGPAT2*, *DGAT2*, *DGKH*, *ENSGALG00000016285*, *LPIN1*, *MBOAT2*, *PNPLA2*
Adipocytokine signaling pathway	6	*CPT1A*, *ENSGALG00000034970*, *LEPR*, *MAPK9*, *PRKAG2*, *SOCS3*

**Table 3 foods-11-02890-t003:** GO terms potentially related to leg muscle quality and meat flavor changes.

Term	Count	Gene List
Muscle alpha-actinin binding	4	*MYPN*, *VCL*, *NRAP*, *PDLIM3*
Glycogen metabolic process	4	*PRKAG2*, *UGP2*, *LEPR*, *PHKA2*
*** Positive regulation of fatty acid oxidation**	3	*LEPR*, *NR4A3*, *C1QTNF2*
Oxygen carrier activity	3	*CYGB*, *HBAD*, *HBA1*
**Positive regulation of triglyceride biosynthetic process**	2	*MFSD2A*, *SLC27A1*
Carbohydrate transport	2	*MFSD2A*, *MFSD2B*
Z disc	6	*MYPN*, *CSRP3*, *VCL*, *NRAP*, *PDLIM3*, *FHL2*
Sarcolemma	5	*SGCD*, *COL6A3*, *VCL*, *COL6A1*, *COL6A2*
**Lipid storage**	3	*PNPLA2*, *NRIP1*, *DGAT2*
**Long-chain fatty acid transport**	2	*MFSD2A*, *FABP3*
Myoblast migration	2	*MEGF10*, *SIX4*
**Triglyceride biosynthetic process**	2	*LPIN1*, *DGAT2*
Carbohydrate binding	6	*COLEC12*, *MAN2A1*, *MRC2*, *GALNT17*, *GALNT14*, *GALNT16*
Collagen binding	5	*COL6A1*, *COL6A2*, *MRC2*, *NID1*, *SMAD3*
Hydrogen peroxide catabolic process	3	*PXDN*, *HBAD*, *HBA1*
Oxygen binding	3	*CYGB*, *HBAD*, *HBA1*
Oxygen transport	3	*CYGB*, *HBAD*, *HBA1*
Peroxidase activity	3	*PXDN*, *HBAD*, *HBA1*
Proteoglycan binding	2	*FN1*, *NID1*
Sarcomere organization	5	*MYPN*, *CSRP3*, *FHOD3*, *WDR1*, *SIX4*

* Bold characters represent significantly enriched GO terms that are related to lipid and fatty acid metabolism.

**Table 4 foods-11-02890-t004:** Metabolites potentially related to flavor compounds and flavor precursors.

Classification	Metabolite List
Lipids and lipid-like molecules	* PC(18:0/20:4(8Z,11Z,14Z,17Z)), LysoPE(22:4(7Z,10Z,13Z,16Z)/0:0), LysoPC(22:4(7Z,10Z,13Z,16Z)), 2-Methylbutyroylcarnitine, Allyl valerate, LysoPE(18:2(9Z,12Z)/0:0), PE-NMe2(9D3/9D3), Phytyl diphosphate, PS(18:4(6Z,9Z,12Z,15Z)/20:0), PC(22:5(7Z,10Z,13Z,16Z,19Z)/P-16:0), Mesterolone, PC(14:0/0:0), Humulol, Glucosylgalactosyl hydroxylysine, Succinylcarnitine, (R)-3-hydroxybutyrylcarnitine, Glycerophosphocholine
Free amino acids and Peptides	D-Ornithine, L-Proline, Cycloposine, Carnosine, Oxidized glutathione
Carbohydrates	N-Acetyl-D-Mannosamine
Vitamins and Cofactors	Pantothenic Acid
Nucleosides, nucleotides, and analogs	DIMP, 5’-CMP, Adenosine 3’,5’-Diphosphate

* PC: phosphatidylcholine; PE: phosphatidyl ethanolamines, PS: phosphatidylserine.

## Data Availability

This study did not report any data.

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
