# Peer review of "Transcriptomic and Metabolomic Profile Analysis of Muscles Reveals Pathways and Biomarkers Involved in Flavor Differences between Caged and Cage-Free Chickens"

_foods, 2022, doi:10.3390/foods11182890_

Round 1

Reviewer 1 Report

A good study to compare the caged and cage free rearing system for poultry. My suggestion to cross check the data regarding pH, WHC, moisture and fat content of meat as these parameters are interlinked but data show different results. Like with pH increase, The WHC increases. Same observation is for collagen content and WHC.

Author Response

Dear reviewers1:

Thank you very much for giving us this opportunity to revise the manuscript. After fully considering the suggestions of the reviewers, following modifications has been made to the manuscript:

  • Introduction: we have reorganized the introduction according to reviewers’ suggestions. Content that was not relevant to our topic was deleted. We added more information about the influence of rearing system and epigenetics to the economic traits according to the recommendations of the reviewer 2.
  • Materials and methods: We added more information to described the meat content analysis and muscle quality assessment. Additionally, the number of individuals used for statical analysis in experiment was added to help readers better understand our works.
  • Results: We deleted some unnecessary description in the results. We rewritten the results of meat content analysis and muscle quality assessment.
  • Discussion: We deleted some duplicate and similar contents. We have added more details about the meat quality related parameters. Additionally, conclusions of the manuscript were also revised.
  • Others: The references have been checked. The missing references were added in the new version. The language of the whole manuscript has been checked by a native English speaker. Abbreviations has been rechecked. The missing institutional email of one of our authors has been added.

We used the revision mode to make revisions easier to be distinguished. If you have any questions, please contact us in time, and we will do our best to meet your requirements. The following are the point-to-point responses to the reviewer:

  • A good study to compare the caged and cage free rearing system for poultry. My suggestion to cross check the data regarding pH, WHC, moisture and fat content of meat as these parameters are interlinked but data show different results. Like with pH increase, The WHC increases. Same observation is for collagen content and WHC.

Answer:  Thank you very much for giving us this opportunity to revise the manuscript. The meat quality related parameter studies included the meat content analysis (using the mixture of boneless breast meat and leg meat) and muscle quality assessment (using the leg and breast muscle, respectively). We have separated these two parts in the result. Interlinked relationships between different parameters do exits. Among which, we are mainly focused on the difference between meat fat content and intramuscular fat, which was one of the most important findings in our result. It has been reported that collagen content is related to WHC and pH, but there was no difference of collagen content in this study. The pH of the samples all varied within the normal range. Additionality, we have summarized and emphasized parameters that conducive to the improvement of meat quality.

We hope the new version will meet your requirements. If you have any questions, please contact us in time. Thanks again.

Reviewer 2 Report

The Manuscript needs minor revision. Please see comments given in the text of reviewed attached file of manuscript.

Author Response

Dear reviewer 2:

Thank you very much for giving us this opportunity to revise the manuscript. After fully considering the suggestions of the reviewers, following modifications has been made to the manuscript:

  • Introduction: we have reorganized the introduction according to reviewers’ suggestions. Content that was not relevant to our topic was deleted. We added more information about the influence of rearing system and epigenetics to the economic traits according to the recommendations of the reviewer 2.
  • Materials and methods: We added more information to described the meat content analysis and muscle quality assessment. Additionally, the number of individuals used for statical analysis in experiment was added to help readers better understand our works.
  • Results: We deleted some unnecessary description in the results. We rewritten the results of meat content analysis and muscle quality assessment.
  • Discussion: We deleted some duplicate and similar contents. We have added more details about the meat quality related parameters. Additionally, conclusions of the manuscript were also revised.
  • Others: The references have been checked. The missing references were added in the new version. The language of the whole manuscript has been checked by a native English speaker. Abbreviations has been rechecked. The missing institutional email of one of our authors has been added.

We used the revision mode to make revisions easier to be distinguished. If you have any questions, please contact us in time, and we will do our best to meet your requirements. The following are the point-to-point responses to the reviewer:

Answer 1: Many thanks for your suggestion. We have added the content in the introduction according to your recommendation.

Answer 2: The language error and missing reference and information from line 141 to 143 has been added in the new version.

Answer 3: The Abbreviations has been checked. Mistake use and missing abbreviations has been corrected in the new version.

Reviewer 3 Report

The article “Transcriptomic and metabolomic profile analysis of muscles reveals pathways and biomarkers involved in flavor differences between caged and cage-free chickens” by Yang et al is of interest. However, the introduction needs to be improved. The introduction should be specific to the topic. I suggest to revise and reorganize the introduction. The authors should clarify the context of the terms "free range" and "cage free". Discussion section also is weak. Please improve and rewrite this section. Besides this, there are additional comments that the authors must address.

Line 34-36: Please provide a reference.

Line 43-44: Please check the sentence for grammatical error.

Lines 96: Please correct m2 as m2.

Lines 116-118: Please replace the word "storage" with the word "stored". Also clarify how long the samples were stored in liquid nitrogen. I presume that samples were snap frozen in liquid nitrogen and transferred to freezer afterwards.

Lines 123: Please remove the word "were".

Lines 131: Please replace the word "to" with the word "of".

Lines 146 & 149: Please check whether the P value was less (<) than 0.05 or higher (>) than 0.05 for significance here and throughout the manuscript.

Lines 188-189: Is this statement relevant to carcass or meat quality?

Lines 200-202: Please remove the repeated statements from these lines.

Lines 213: “because the females were more sensitive to the flavor.” Please provide a reference for the statement.

Lines 237 & 273: Are these statements describing the results?

Line 307-311: This information is also available in the introduction section. Please avoid repetition.

Line 312-314: This information is also repeating in the methodology.

Line 403-411: Conclusions should be specific to the study. Please improve the conclusions with special reference to the futuristic studies.

Author Response

Dear reviewers 3:

Thank you very much for giving us this opportunity to revise the manuscript. After fully considering the suggestions of the reviewers, following modifications has been made to the manuscript:

  • Introduction: we have reorganized the introduction according to reviewers’ suggestions. Content that was not relevant to our topic was deleted. We added more information about the influence of rearing system and epigenetics to the economic traits according to the recommendations of the reviewer 2.
  • Materials and methods: We added more information to described the meat content analysis and muscle quality assessment. Additionally, the number of individuals used for statical analysis in experiment was added to help readers better understand our works.
  • Results: We deleted some unnecessary description in the results. We rewritten the results of meat content analysis and muscle quality assessment.
  • Discussion: We deleted some duplicate and similar contents. We have added more details about the meat quality related parameters. Additionally, conclusions of the manuscript were also revised.
  • Others: The references have been checked. The missing references were added in the new version. The language of the whole manuscript has been checked by a native English speaker. Abbreviations has been rechecked. The missing institutional email of one of our authors has been added.

We used the revision mode to make revisions easier to be distinguished. If you have any questions, please contact us in time, and we will do our best to meet your requirements. The following are the point-to-point responses to the reviewer:

  • The article “Transcriptomic and metabolomic profile analysis of muscles reveals pathways and biomarkers involved in flavor differences between caged and cage-free chickens” by Yang et al is of interest. However, the introduction needs to be improved. The introduction should be specific to the topic. I suggest to revise and reorganize the introduction. The authors should clarify the context of the terms "free range" and "cage free". Discussion section also is weak. Please improve and rewrite this section. Besides this, there are additional comments that the authors must address.

Answer: Many thanks for your suggestion. We have clarified the context of the terms "free range" and "cage free" in introduction. We also reorganized the introduction and discussion in the new version. We hope the new version will meet your requirements. If you have any questions, please contact us in time.

  • Line 34-36: Please provide a reference.

Answer:Thanks for your reminder. The reference has been added.

  • Line 43-44: Please check the sentence for grammatical error.

Answer: We have corrected the grammatical error.

  • Lines 96: Please correct m2 as m2.

Answer: Thanks for your reminder. We have corrected it in the new version.

  • Lines 116-118: Please replace the word "storage" with the word "stored". Also clarify how long the samples were stored in liquid nitrogen. I presume that samples were snap frozen in liquid nitrogen and transferred to freezer afterwards.

Answer: Thanks for your reminder. We have replaced the word "storage" with the word "stored". In our lab, important samples were usually stored in the liquid nitrogen until use. Since the temperature of liquid nitrogen is lower, it is more conducive to the preservation of samples.

  • Lines 123: Please remove the word "were".

Answer: Many thanks for your reminder. We have corrected it in the new version.

  • Lines 131: Please replace the word "to" with the word "of".

Answer: Many thanks for your reminder. We have corrected it in the new version.

  • Lines 146 & 149: Please check whether the P value was less (<) than 0.05 or higher (>) than 0.05 for significance here and throughout the manuscript.

Answer: Many thanks for your reminder. The P value is used to measure the probability of making mistakes in statistical analysis. The lower the p value, the less possibility to make mistakes. We selected the P value less than 0.05 as the threshold for significance in this study.

  • Lines 188-189: Is this statement relevant to carcass or meat quality?

Answer: Many thanks for your reminder. We have deleted this unnecessary sentence in in the new version.

  • Lines 200-202: Please remove the repeated statements from these lines.

Answer: Many thanks for your reminder. We have rewritten these sentences to avoid repeated in the new version.

  • Lines 213: “because the females were more sensitive to the flavor.” Please provide a reference for the statement.

Answer: Thanks for your suggestion, we have added a reference in this statement.

  • Lines 237 & 273: Are these statements describing the results?

Answer: Many thanks for your suggestion. These statements are trying to tell readers that data (RNA-seq data and metabolomic data) used for following analysis were good enough in quality and quantity, which was a part of data pre-analysis. We have revised these sentences to make them more easily to be understood.

  • Line 307-311: This information is also available in the introduction section. Please avoid repetition.

Answer: Thanks for your suggestion, we have deleted it in the new version.

  • Line 312-314: This information is also repeating in the methodology.

Answer: Thanks for your suggestion, we have revised it in the new version.

  • Line 403-411: Conclusions should be specific to the study. Please improve the conclusions with special reference to the futuristic studies.

Answer: Many thanks for your suggestion, we have rewritten the conclusions in the new version.

Round 2

Reviewer 3 Report

.